# B cell analyses after SARS-CoV-2 mRNA third vaccination reveals a hybrid immunity like antibody response

Emanuele Andreano [1,11], Ida Paciello[1,11], Giulio Pierleoni [2], Giulia Piccini [3], Valentina Abbiento[1], Giada Antonelli[1], Piero Pileri [1], Noemi Manganaro [1], Elisa Pantano[1], Giuseppe Maccari [4], Silvia Marchese [5], Lorena Donnici [6], Linda Benincasa[2], Ginevra Giglioli[2], Margherita Leonardi[2,3], Concetta De Santi[1], Massimiliano Fabbiani[7], Ilaria Rancan[7], Mario Tumbarello[7,8], Francesca Montagnani [7,8], Claudia Sala[1], Duccio Medini[4], Raffaele De Francesco [5,6], Emanuele Montomoli[2,3,9] & Rino Rappuoli [1,10] ✉

The continuous evolution of SARS-CoV-2 generated highly mutated variants able to escape natural and vaccine-induced primary immunity. The administration of a third mRNA vaccine dose induces a secondary response with increased protection. Here we investigate the longitudinal evolution of the neutralizing antibody response in four donors after three mRNA doses at single-cell level. We sorted 4100 spike protein specific memory B cells identifying 350 neutralizing antibodies. The third dose increases the antibody neutralization potency and breadth against all SARS-CoV-2 variants as observed with hybrid immunity. However, the B cell repertoire generating this response is different. The increases of neutralizing antibody responses is largely due to the expansion of B cell germlines poorly represented after two doses, and the reduction of germlines predominant after primary immunization. Our data show that different immunization regimens induce specific molecular signatures which should be considered while designing new vaccines and immunization strategies.

The emergence of SARS-CoV-2 variants of concern (VoCs) able to escape vaccine immunity elicited by the spike (S) protein of the original virus isolated in Wuhan, China, have decreased the impact of vaccination[1,2]. As a consequence, the COVID-19 pandemic continues to impact the health, the economy, and the freedom of people worldwide in spite of the several billion doses of vaccines already deployed. This scenario raises important questions about the use of the existing vaccines and the immunity they can provide to tackle current and future variants. Therefore, it became of utmost importance to understand the nature and the quality of the immune response elicited by the different vaccines used worldwide. In our previous study, we analyzed at a single-cell level the immune response induced by two doses of the BNT162b2 mRNA vaccine in naïve people and in people that had been previously infected by the SARS-CoV-2 virus[3]. In this study, we analyzed at a single-cell level the longitudinal B cell and neutralizing antibody response of the same

[1]Monoclonal Antibody Discovery (MAD) Lab, Fondazione Toscana Life Sciences, Siena, Italy. [2]VisMederi Research S.r.l., Siena, Italy. [3]VisMederi S.r.l, Siena, Italy. [4]Data Science for Health (DaScH) Lab, Fondazione Toscana Life Sciences, Siena, Italy. [5]Department of Pharmacological and Biomolecular Sciences DiSFeB, University of Milan, Milan, Italy. [6]INGM, Istituto Nazionale Genetica Molecolare "Romeo ed Enrica Invernizzi", Milan, Italy. [7]Department of Medical Sciences, Infectious and Tropical Diseases Unit, Siena University Hospital, Siena, Italy. [8]Department of Medical Biotechnologies, University of Siena, Siena, Italy. [9]Department of Molecular and Developmental Medicine, University of Siena, Siena, Italy. [10]Department of Biotechnology, Chemistry and Pharmacy, University of Siena, Siena, Italy. [11]These authors contributed equally: Emanuele Andreano, Ida Paciello. ✉e-mail: rino.rappuoli@biotecnopolo.it

naïve people after a third immunization. We found that, while the overall immune response after a third dose is similar to that observed in the hybrid immunity of vaccinated people previously infected by the virus, the B cell repertoire that stands behind the response is different.

## Results

### B cells response after the third dose

To evaluate the longitudinal evolution of the neutralizing antibody response, four seronegative donors that participated in our previous study after two doses of BNT162b2 mRNA vaccine[3], were re-enrolled after receiving the third dose. None of the subjects were exposed to SARS-CoV-2 infection between the second and third vaccination dose. Data after the third dose (seronegative third dose; SN3) described in this study were compared to those obtained from the same subjects after the second dose (seronegative second dose; SN2) and to those of subjects with hybrid immunity (seropositive second dose; SP2) previously described[3]. Three subjects received the BNT162b2 (VAC-001, VAC-002, and VAC-008) vaccine, while one subject (VAC-010) received the mRNA-1273 vaccine. Blood collection occurred at an average of 58 days post third vaccination dose. Subject details are summarized in Supplementary Table 1 and the gating strategy used is shown in Supplementary Fig. 1a. The frequency of $CD19^+CD27^+IgD^-IgM^-$ memory B cell (MBCs) specific for the S protein$^+$ was 6.18-fold higher in SN3 compared to SN2 (Fig. 1a, b). This was also 2.26-fold higher than that previously observed in subjects with SARS-CoV-2 infection and subsequent vaccination with two doses of the BNT162b2 mRNA vaccine (SP2)[3]. No major differences between the two groups were observed in the $CD19^+CD27^+IgD^-IgM^-/IgM^+$ memory B cell (MBCs) compartments (Supplementary Fig. 1b–e). In addition, SN3 showed higher binding to the S protein, receptor binding domain (RBD), and N-terminal domain (NTD), and a 5.82-fold higher neutralization activity against the original Wuhan SARS-CoV-2 virus compared to SN2 (Fig. 1c, d and Supplementary Fig. 1f–i).

### Boosting antibody potency and breadth

To evaluate the longitudinal B cell response, antigen-specific class-switched MBCs ($CD19^+CD27^+IgD^-IgM^-$) were single-cell sorted using as bait the Wuhan prefusion SARS-CoV-2 S protein trimeric antigen, which was encoded by the mRNA vaccine. Sorted cells were incubated for 2 weeks to naturally release human monoclonal antibodies (mAbs) into the supernatant. A total of 4100 S protein$^+$ MBCs were sorted and 2436 (59.4%) produced mAbs able to recognize the S protein prefusion trimer in ELISA. Of these, 350 neutralized the original Wuhan live SARS-CoV-2 virus when tested at a single point dilution (1:10) by cytopathic effect-based microneutralization assay (CPE-MN). Overall, the fraction of S protein-specific B cells producing neutralizing antibodies (nAbs) was 14.4% which is 2.2-fold higher than what was observed in SN2 and comparable to what was observed for SP2 dose vaccinees (14.8%) in our previous study (Supplementary Fig. 2a, b and Supplementary Table 2)[3]. To better characterize identified nAbs, we tried to express all 350 as immunoglobulin G1 (IgG1), and we were able to recover and express 206 of them. Binding by ELISA to RBD and NTD of the original Wuhan SARS-CoV-2 S had similar frequency in SN2 and SN3, while we observed a reduction of S protein trimer-specific antibodies after a third booster dose, a trend similar to what was observed in SP2 (Supplementary Fig. 2c). No S2 domain binding nAbs were identified. Finally, we evaluated the neutralization potency against the original Wuhan virus, the alpha (B.1.1.7), beta (B.1.351), gamma (P.1) delta (1.617.2), omicron BA.1 (B.1.1.529.1), and omicron BA.2 (B.1.1.529.2) and compared it with SN2 (Fig. 1e–l). To increase the power of the analyses, in Fig. 1, we included all 52 nAbs isolated from the five seronegative donors enrolled in our previous study[3]. Overall, nAbs isolated in SN3 were higher in frequency and potency against all tested viruses compared to SN2 and showed a significantly higher 100% inhibitory

concentration ($IC_{100}$), except for BA.1 and BA.2 VoCs, where SN2 did not have enough nAbs to make a meaningful comparison (Fig. 1e–l). The $IC_{100}$ geometric mean (GM-$IC_{100}$) in SN3 was 3.25, 7.57-, 3.31-, 3.21-, 5.70-, and 2.83-fold lower compared to SN2 for the Wuhan virus, the alpha, beta, gamma, delta, and BA.1 VoCs, respectively (Fig. 1 and Supplementary Fig. 3). Only for BA.2 a higher GM-$IC_{100}$ was observed in SN2 compared to SN3. Interestingly, SN3 also showed a higher neutralization potency compared to SP2 against all tested viruses with a 1.13-, 3.49-, 2.81, 2.23-, 3.15-, 2.22-, and 1.20-fold lower GM-$IC_{100}$ (Fig. 1l and Supplementary Fig. 3). In addition, we observed that nAbs induced following a third booster dose retained a high ability to cross-neutralize all tested SARS-CoV-2 VoCs. Indeed, while 67.3, 21.1, 25.0, 51.9, 1.9, and 7.7% of nAbs maintained the ability to neutralize alpha, beta, gamma, delta, omicron BA.1, and omicron BA.2, respectively, in the SN2 group[3], the same viruses were neutralized by 77.2, 41.7, 39.3, 61.2, 21.8, and 25.2% of nAbs in the SN3 group (Fig. 1 and Supplementary Table 2). Finally, we assessed if a third booster dose would enhance the ability of nAbs to recognize and cross-neutralize the distantly related SARS-CoV-1 virus. Two of the four SN3 donors presented nAbs able to recognize SARS-CoV-1 (Supplementary Fig. 4a). However, only 2.4% (5/206) and 0.9% (2/206) of all nAbs were able to bind the SARS-CoV-1 S protein and neutralize this sarbecovirus respectively, showing a lower frequency than the one observed in SN2 and SP2 nAbs (Supplementary Fig. 4a, b)[3].

### Antibody classes of cross-protection

To understand the epitope region recognized by our RBD targeting antibodies ($n = 154$) after a third vaccination dose, we performed a competition assay with three known antibodies. As previously described, we used the Class 1/2 antibody J08[4], the Class 3 antibody S309[5], and the Class 4 antibody CR3022[6] to map the epitope regions targeted by our RBD-binding nAbs[3]. The most abundant class of nAbs targeted the Class 1/2 epitope region (98/154; 63.6%), followed by Class 3 targeting nAbs (30/154; 19.5%) and nAbs that recognize the Class 4 region (3/154; 1.9%). The remaining 23 (14.9%) nAbs did not compete with any of the three antibodies used in our competition assay. As shown in Fig. 2a, in SN3, we observed a marked increase in Class 3 and 4 nAbs compared to SN2 and a similar distribution to SP2. When we compared the GM-$IC_{100}$ of classes of nAbs in the SN3 with the SN2 and SP2 groups, we observed similar neutralization potency when tested against the SARS-CoV-2 virus originally isolated in Wuhan, China (Supplementary Fig. 5). Following, we aimed to understand which classes of RBD ($n = 154$) and NTD ($n = 43$) targeting nAbs in SN3 were mainly responsible for the cross-protection against the SARS-CoV-2 VoCs. Overall, we observed that Class 1/2 nAbs are the most abundant family of antibodies against all tested variants, followed by Class 3, Not-competing, NTD, and Class 4 nAbs (Fig. 2b, c). Finally, we compared the functional antibody response between SN3 and SP2 against the highly mutated omicron BA.1 and BA.2 viruses. Interestingly, we observed that a third mRNA booster dose increases neutralization potency and evasion resistance of Class 1/2 nAbs against both omicron BA.1 and BA.2 compared to SP2. The opposite trend was observed for Class 3 nAbs, which had a lower frequency of cross-protection in SN3 compared to SP2 (Fig. 2d, e).

### Antibody gene repertoire

We then interrogated the functional antibody repertoire. Initially, we analyzed all immunoglobulin heavy chain sequences retrieved from the three different groups (SN2 $n = 58$; SN3 $n = 288$; SP2 $n = 278$), and their respective V-J gene rearrangements (IGHV;IGHJ)[3]. Interestingly, SN2 and SN3 share only 20.2% percent (23/114) of SN3 IGHV;IGHJ rearrangements, while SN3 and SP2 share up to 45.6% (52/114). In addition, we observed that the frequency of antibodies encoded by predominant rearrangements induced by two vaccination doses (i.e. IGHV3-30;IGHJ6-1, IGHV3-33;IGHJ4-1, IGHV3-53;IGHJ6-1, and IGHV3-

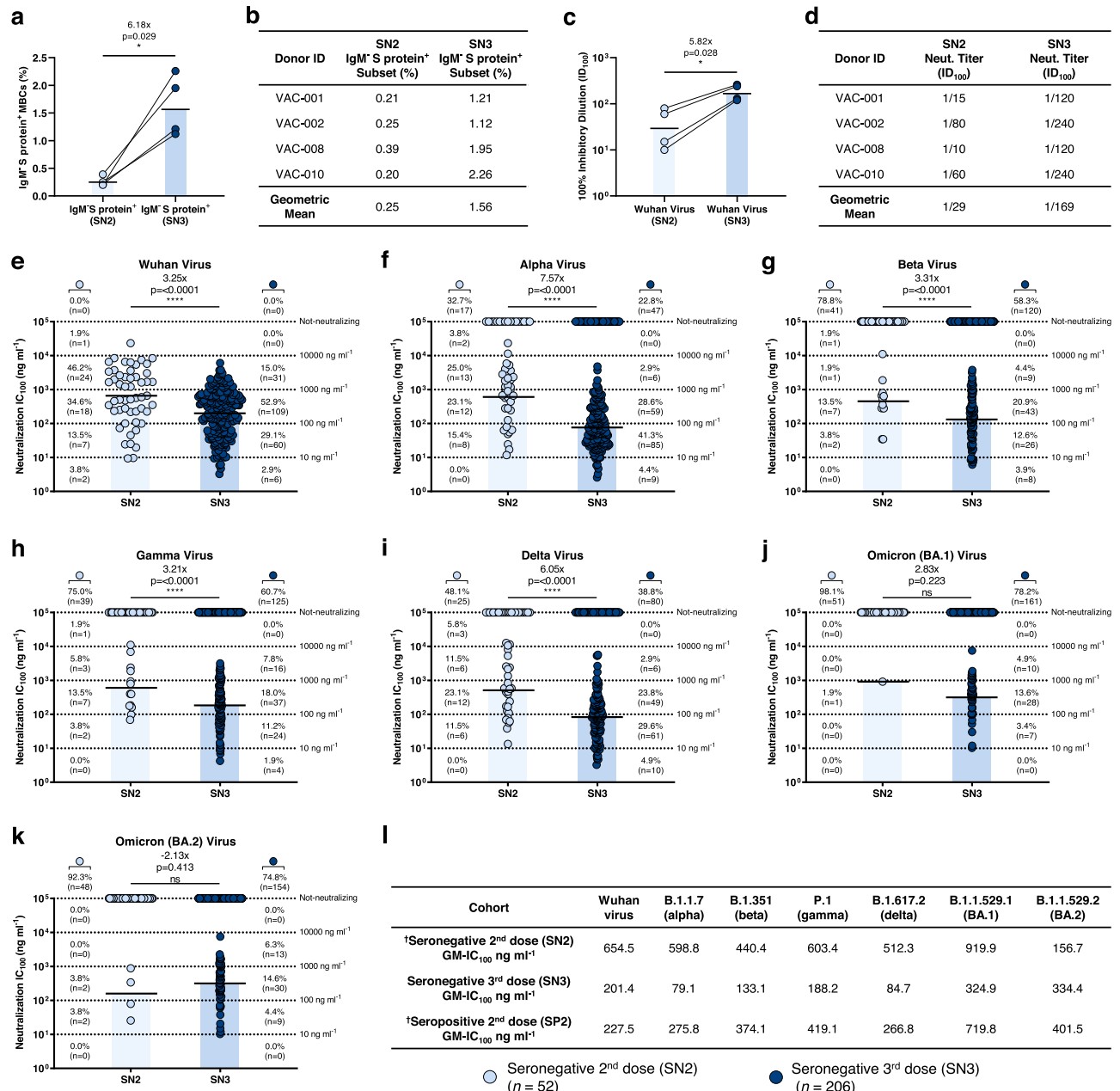

**Fig. 1 | Potency and breadth of neutralization of nAbs against SARS-CoV-2 and VoCs. a** The graph shows the frequency of CD19+CD27+IgD−IgM− able to bind the SARS-CoV-2 S protein trimer (S protein+) in SN2 and SN3 donors (n = 4/group). The black line and bars denote the geometric mean. **b** The table summarizes the frequencies of the B cell populations in SN2 and SN3. **c** The graph shows the neutralizing activity of plasma samples against the original Wuhan SARS-CoV-2 virus for SN2 (n = 4) and SN3 (n = 4). Technical duplicates were performed for each experiment. **d** The table summarizes the 100% inhibitory dilution (ID$_{100}$) of each COVID-19 vaccinee and the geometric mean for SN2 and SN3 (n = 4/group). **e**–**k** Scatter dot charts show the neutralization potency, reported as IC$_{100}$ (ng ml$^{-1}$), of nAbs tested against the original Wuhan SARS-CoV-2 virus (**e**) and the alpha (**f**), beta (**g**), gamma (**h**), delta (**i**), omicron BA.1 (**j**), and omicron BA.2 (**k**) VoCs. The number of samples tested (n) and percentage of nAbs from SN2 vs SN3, fold change, and statistical significance are denoted on each graph. **l** The table shows the IC$_{100}$ geometric mean (GM) of all nAbs pulled together from SN2, SN3, and SP2 against all SARS-CoV-2 viruses tested. "†" indicates previously published data[3]. Technical duplicates were performed for each experiment. A nonparametric Mann–Whitney t-test was used to evaluate the statistical significance between groups. Two-tailed p value significances are shown as *p < 0.05, **p < 0.01, ***p < 0.001, and ****p < 0.0001. Source data are provided as a Source Data file.

66;IGHJ4-1)[3,4,7,8] were reduced after a third booster dose, while we observed an expansion of the antibody germlines IGHV1-58;IGHJ3-1 and IGHV1-69;IGHJ4-1 which were previously found to be predominant in SP2[3,9] (Fig. 3a and Supplementary Fig. 6). These latter germlines previously showed a high level of cross-neutralization activity against SARS-CoV-2 VoCs[10-12]. In addition, we observed in one donor (VAC-001) an important expansion of the germline IGHV1-46;IGHJ6-1 after receiving a third booster dose (Fig. 3a and Supplementary Fig. 6).

Conversely to what found in SP2[3], we did not observe the expansion of Class 3 targeting antibody germline IGHV2-5;IGHJ4-1, which so far has been observed only in previously infected vaccinees or subjects immunized with adenoviral vectors[3,13,14]. Following this, we aimed to identify expanded clonal families within the same four donors after a second and third vaccination dose. Sequences (SN2 n = 43; SN3 n = 288) were clustered by binning the clones to their inferred germlines (centroids) and according to 80% nucleotide sequence in the

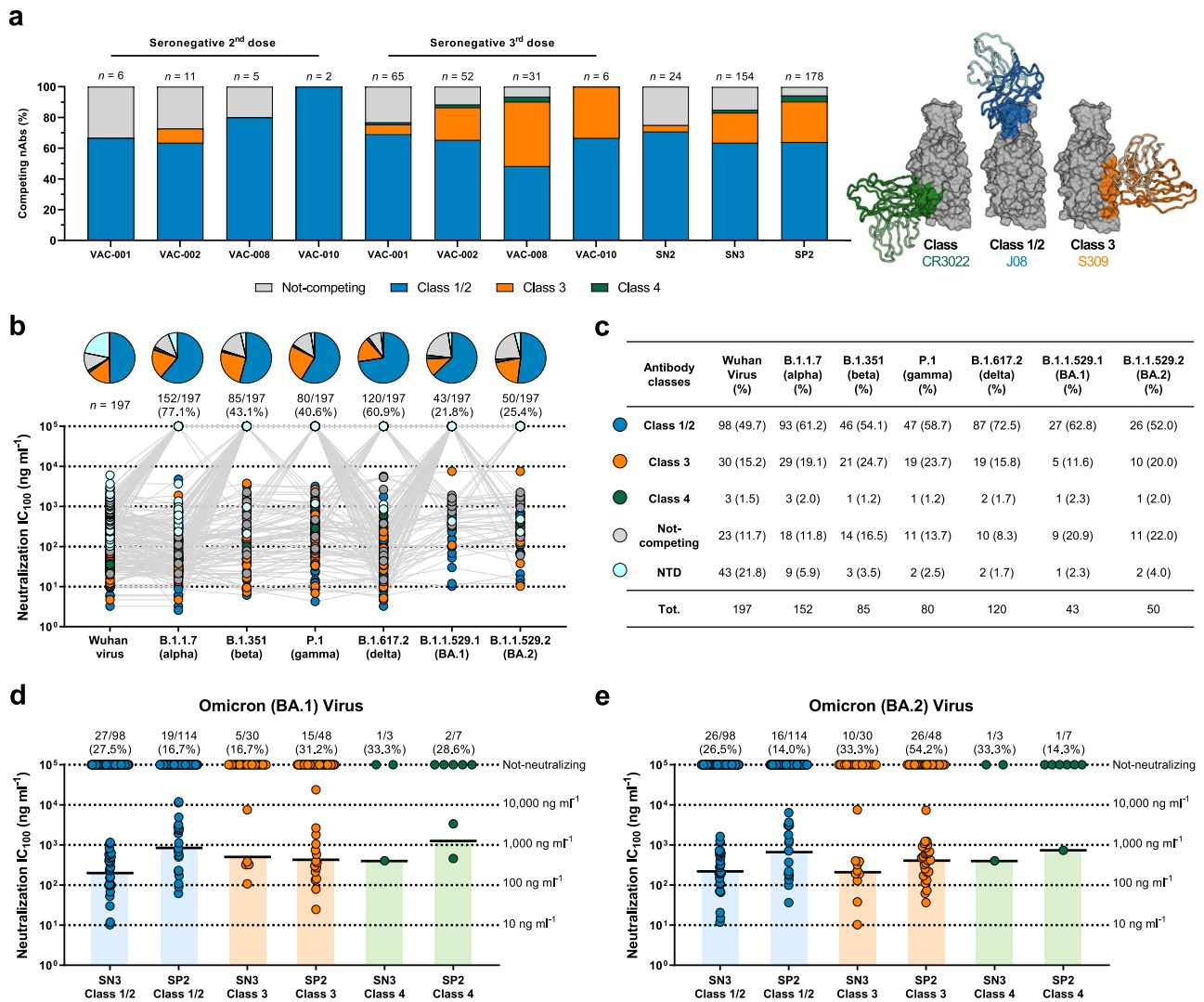

**Fig. 2 | Distribution of SN3 nAbs against SARS-CoV-2 VoCs. a** The bar graph shows the epitope regions recognized by RBD-binding nAbs. Class 1/2, Class 3, Class 4, and not-competing nAbs are shown in light blue, orange, green, and light gray, respectively. The number (*n*) of antibodies tested per each donor is denoted on the graph. The right panel highlights the epitope recognized by Class 1/2 (J08), Class 3 (S309) and Class 4 (CR3022) antibodies used in our competition assay. **b** Pie charts show the distribution of cross-protective nAbs based on their ability to bind Class 1/2, Class 3, and Class 4 regions on the RBD, as well as not-competing nAbs (gray) and NTD-targeting, nAbs (cyan). Dot charts show the neutralization potency, reported

as IC$_{100}$ (ng ml$^{-1}$), of nAbs against the Wuhan virus and alpha, beta, gamma, delta, omicron BA.1, and omicron BA.2 SRS-CoV-2 VoCs observed in the SN3 group. The number and percentage of nAbs are denoted on each graph. **c** the table summarizes the number and percentages of Class 1/2, Class 3, Class 4, not-competing, and NTD-targeting nAbs for each tested variant. **d**, **e** Dot charts compare the distribution of nAbs between SN3 and SP2 groups against omicron BA.1 (**d**) and BA.2 (**e**). The number, percentage and GM-IC$_{100}$ (black lines and colored bars) of nAbs are denoted on each graph. Source data are provided as a Source Data file.

heavy complementary determining region 3 (CDRH3). Clusters were defined as antibody families including at least five or more members as previously described[15]. Of the 331 sequences, 226 (68.3%) were orphans (i.e., did not cluster with other sequences), and only in six cases sequences from SN2 and SN3 were binned to the same centroid (Fig. 3b). Only five clusters were identified, three of which were composed by antibodies belonging exclusively from the SN3 group. Of these clusters, two were formed by antibody sequences shared between the SN2 and SN3 groups. The smallest cluster was composed of 11 antibody members encoded from the IGHV1-58;IGHJ3-1 germline, while the biggest cluster, composed of 18 antibody members, were encoded by the IGHV3-48/IGHV3-53/IGHV3-66 germlines (Fig. 3b). Finally, we evaluated the V gene mutation levels, neutralization potency and breadth in nAbs encoded by reduced and expanded germlines following a third vaccination dose. Our data showed that IGHV3-53;IGHJ6-1 and IGHV3-66;IGHJ4-1 germlines, expanded in SN2 and reduced in SN3, have a similar level of V gene somatic mutations,

are poorly cross-reactive, especially against omicron BA.1 and BA.2, and show medium neutralization potency (IC$_{100}$ mainly between 100 and 1000 ng ml$^{-1}$). Differently, the antibody germlines IGHV1-58;IGHJ3-1 and IGHV1-69;IGHJ4-1, mainly expanded in SN3 and poorly represented in SN2, show between 5 and 15-fold higher V gene somatic mutation levels, higher neutralization potency and breadth against all SARS-CoV-2 VoCs (Fig. 3c–f).

## Discussion

In agreement with other longitudinal studies[7,8,16], we found that the third dose of mRNA vaccine induces an immune response similar to the hybrid immunity observed in people vaccinated after SARS-CoV-2 infection. This antibody response is characterized by a small increase in S protein binding antibodies, a strong increase in neutralizing potency, and a considerable increase in antibodies able to cross-neutralize emerging variants, including omicron BA.1 and BA.2. The increased potency and breadth are due to a significant expansion of S

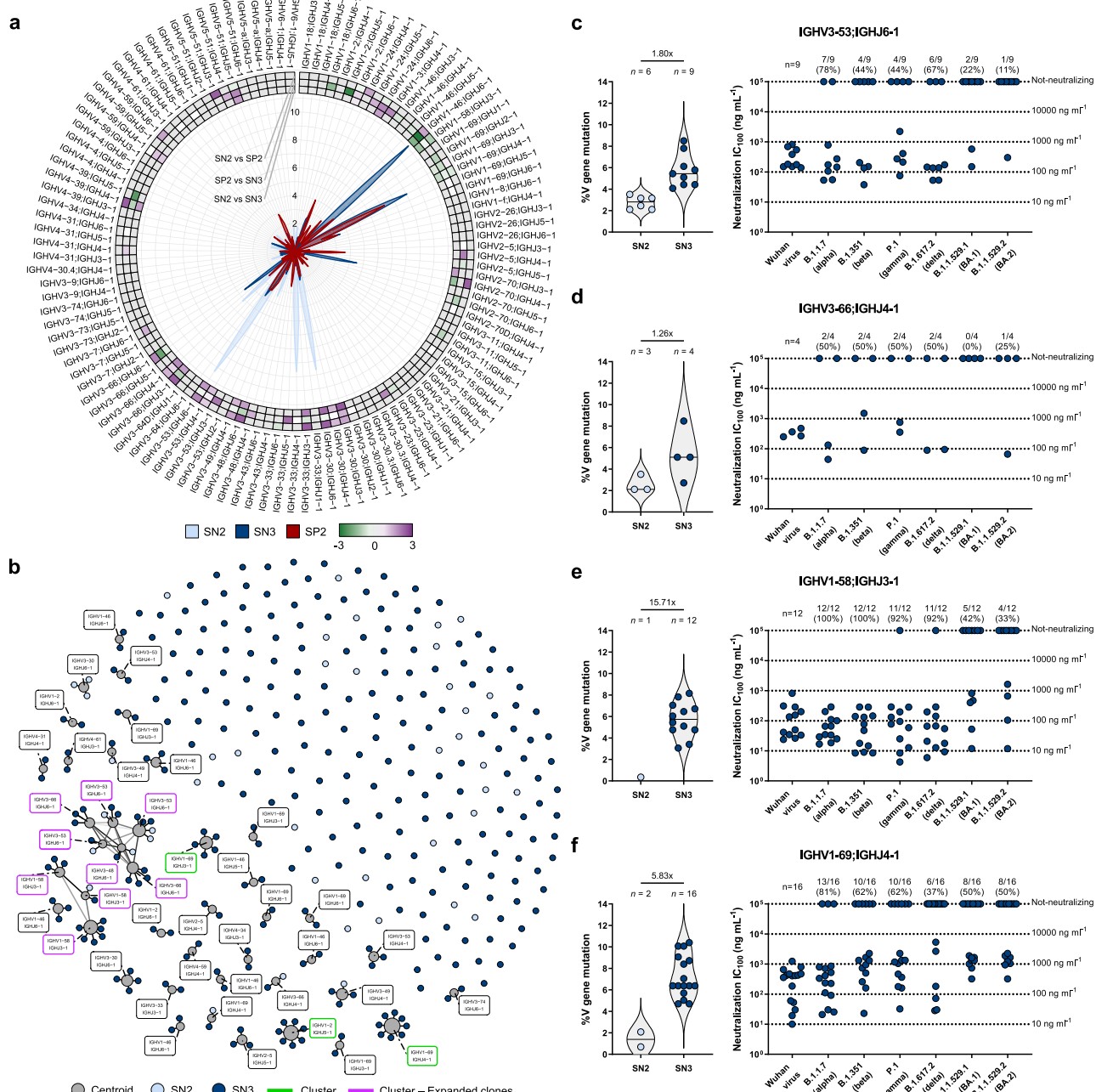

**Fig. 3 | Repertoire analyses and characterization of predominant antibody germlines. a** Radar plot shows the distribution of IGHV;IGHJ germlines among the three different groups. SN2, SN3, and SP2 are shown in light blue, dark blue, and red, respectively. Heatmap represents the Log2 fold change (FC) among groups. **b** Network plot shows the clonally-expanded antibody families in SN2 and SN3. Centroids and nAbs from SN2 and SN3 groups are shown in gray, light blue, and dark blue, respectively. Clusters and expanded clones are highlighted in bright green and bright purple, respectively. **c–f** The graphs show the V gene somatic mutation frequency (left panel) and neutralization potency (IC$_{100}$; right panel) of IGHV3-53;IGHJ6-1 (**c**), IGHV3-66;IGHJ4-1 (**d**), IGHV1-58;IGHJ3-1 (**e**), and IGHV1-69;IGHJ4-1 (**f**) gene derived nAbs, against the original SARS-CoV-2 virus first detected in Wuhan, China, and all SARS-CoV-2 VoCs. The number and percentage of nAbs analyzed are denoted on each graph. Source data are provided as a Source Data file.

protein-specific MBCs, which is even higher than that observed in subjects with hybrid immunity, and by a strong increase of V gene somatic mutations. What is interesting in our study is that the increased potency and breadth observed after a third booster dose was due mostly to Class 1/2 nAbs, while Class 3 antibodies had a lower frequency and breadth compared to subjects with hybrid immunity. In addition, we found that a third mRNA dose did not induce a strong response against the distantly related SARS-CoV-1, suggesting that additional doses of homologous vaccines against SARS-CoV-2 will focus the antibody response against this virus instead of broadening

cross-protection to other coronaviruses. Another important observation of this study is that the increased neutralization potency and breadth is not due to a linear evolution of the B cells producing nAbs after two vaccine doses but is due mostly to the expansion of new B cells, which were not detected after primary immunization. Indeed, the secondary response induced by a third vaccine dose did not derive from expanded B cell clones, but it was dominated by singlets that constituted 68% of the entire repertoire. A possible explanation for this scenario is that newly expanded B cell clones needed further maturation, induced by a third booster dose, before acquiring a high level of

functionality and being selected for further expansion. In addition, the germlines IGHV3-53;IGHJ6-1/IGHV3-66;IGHJ4-1 dominated the neutralizing response in donors infected with the original Wuhan virus and in subjects immunized with two doses of mRNA vaccines[3,4,17–19], decreased in frequency and did not improve in potency or cross-neutralization after a third dose. Conversely, the antibody germlines IGHV1-58;IGHJ3-1 and IGHV1-69;IGHJ4-1 became largely responsible for the improved potency and cross-neutralization observed after a third dose. Interestingly, the IGHV1-58 germline predominant in this group was previously shown to recognize a "supersite" on the S protein surface and to be expanded following beta or omicron breakthrough infection in vaccinated individuals[10,12,20]. An interesting finding of this work is that the highly cross-reactive germline IGHV2-5;IGHJ4-1, found in subjects with hybrid immunity and in people vaccinated with viral vectors[3,14], is absent after a third mRNA vaccine, suggesting that the induction of this germline may require endogenous production of the S protein. The lack of expansion of the IGHV2-5;IGHJ4-1 germline after three mRNA vaccine doses could partially explain the above-mentioned loss of cross-reactivity by Class 3 nAbs. Overall, our study provides a high-resolution picture of the functional and genetic properties of a third mRNA vaccination and, despite we observed important similarities with hybrid immunity, unravels features of the antibody response specifically produced after a third mRNA vaccine dose. These observations are very important and should be considered while designing new vaccines and implementing vaccination regimens for booster doses, especially in low-middle-income countries where less than 20% of people received at least one vaccine dose[21].

## Methods

### Enrollment of COVID-19 vaccinees and human sample collection
This work results from a collaboration with the Azienda Ospedaliera Universitaria Senese, Siena (IT) that provided samples from COVID-19 vaccinated donors, of both sexes, who gave their written consent. The study was approved by the Comitato Etico di Area Vasta Sud Est (CEAVSE) ethics committees (Parere 17065 in Siena) and conducted according to good clinical practice in accordance with the declaration of Helsinki (European Council 2001, US Code of Federal Regulations, ICH 1997). This study was unblinded and not randomized. No statistical methods were used to predetermine the sample size.

### Single-cell sorting of SARS-CoV-2 S protein+ memory B cells from COVID-19 vaccinees
Peripheral blood mononuclear cells (PBMCs) and single-cell sorting strategy were performed as previously described[3,4]. Briefly, PBMC were isolated from heparin-treated whole blood by density gradient centrifugation (Ficoll-Paque™ PREMIUM, Sigma-Aldrich) and stained with Live/Dead Fixable Aqua (Invitrogen; Thermo Scientific) diluted 1:500. After 20 min incubation cells were saturated with 20% normal rabbit serum (Life Technologies) for 20 min at 4 °C and then stained with SARS-CoV-2 S protein labeled with Strep-Tactin®XT DY-488 (IBA-Lifesciences cat# 2-1562-050) for 30 min at 4 °C. The SARS-CoV-2 S protein used for single-cell sorting was generated using the plasmid SARS-CoV-2 S-2P which contains two consecutive proline substitutions in the S2 subunit, HRV 3 C cleavage site (before tags), 8X His-tag and 2X Strep-Tag II on the C-terminal backbone[22,23]. After incubation, the following staining mix was used CD19 V421 (BD cat# 562440, 1:320), IgM PerCP-Cy5.5 (BD cat# 561285, 1:50), CD27 PE (BD cat# 340425, 1:30), IgD-A700 (BD cat# 561302, 1:15), CD3 PE-Cy7 (BioLegend cat# 300420, 1:100), CD14 PE-Cy7 (BioLegend cat# 301814, 1:320), CD56 PE-Cy7 (BioLegend cat# 318318, 1:80), and cells were incubated at 4 °C for additional 30 min. Stained MBCs were gated as live/dead, morphology, CD19+CD3−CD14−CD56−, CD27+IgD−, IgM−, and S protein+. Single-cell sorting was performed with a BD FACS Aria III (BD Biosciences) into 384-well plates containing 3T3-CD40L feeder cells (NIH AIDS Reagent Program, Cat#12535), IL-2 and IL-21 and incubated for 14 days as previously described[24].

### ELISA assay with SARS-CoV-2 and SARS-CoV-1 S protein prefusion trimer
mAbs and plasma binding specificity against the S protein trimer was detected by ELISA as previously described[3]. Briefly, 384-well plates (microplate clear, Greiner Bio-one) were coated with 3 µg/mL of streptavidin (Thermo Fisher) diluted in carbonate-bicarbonate buffer (E107, Bethyl laboratories) and incubated at RT overnight. The next day, plates were incubated for 1 h at RT with 3 µg/mL of SARS-CoV-2 or SARS-CoV-1 S protein, and saturated with 50 µL/well of blocking buffer (phosphate-buffered saline, 1% BSA) for 1 h at 37 °C. Following, 25 µL/well of mAbs or plasma samples, diluted 1:5 or 1:10, respectively, in sample buffer (phosphate-buffered saline, 1% BSA, 0.05% Tween-20), were added serially diluted step dilution 1:2 and then incubated at 1 h at 37 °C. Finally, 25 µL/well of alkaline phosphatase-conjugated goat antihuman IgG and IgA (Southern Biotech) diluted 1:2000 in sample buffer were added. S protein binding was detected using 25 µL/well of PNPP (p-nitrophenyl phosphate; Thermo Fisher) and the reaction was measured at a wavelength of 405 nm by the Varioskan Lux Reader using the SkanIt Software Microplate Readers 6.0.1 (Thermo Fisher Scientific). After each incubation step, plates were washed three times with 100 µL/well of washing buffer (phosphate-buffered saline, 0.05% Tween-20). A sample buffer was used as a blank and the threshold for sample positivity was set at twofold the optical density (OD) of the blank. Technical duplicates were performed for mAbs and technical triplicates were performed for sera samples.

### ELISA assay with RBD, NTD, and S2 subunits
mAbs identification and plasma screening of vaccinees against RBD, NTD, or S2 SARS-CoV-2 protein were performed by ELISA as previously described[3]. Briefly, 3 µg/mL of RBD, NTD, or S2 SARS-CoV-2 protein diluted in carbonate-bicarbonate buffer (E107, Bethyl laboratories) were coated in 384-well plates (microplate clear, Greiner Bio-one) and blocked with 50 µL/well of blocking buffer (phosphate-buffered saline, 1% BSA) for 1 h at 37 °C. After washing, plates were incubated for 1 h at 37 °C with mAbs diluted 1:5 in samples buffer (phosphate-buffered saline, 1% BSA, 0.05% Tween-20) or with plasma at a starting dilution 1:10 and step diluted 1:2 in sample buffer. Anti-Human IgG −Peroxidase antibody (Fab specific) produced in goat (Sigma) diluted 1:45,000 in sample buffer was then added and samples incubated for 1 h at 37 °C. Plates were then washed and incubated with TMB substrate (Sigma) for 15 min before adding the stop solution (H$_2$SO$_4$ 0.2 M). The OD values were identified using the Varioskan Lux Reader (Thermo Fisher Scientific) at 450 nm. Each condition was tested in triplicate and samples tested were considered positive if the OD value was twofold the blank.

### Flow cytometry-based competition assay
To classify mAbs candidates on the basis of their interaction with Spike epitopes, we performed a flow cytometry-based competition assay as previously described[3,9]. Briefly, magnetic beads (Dynabeads His-Tag, Invitrogen) were coated with histidine-tagged S protein according to the manufacturer's instructions. Then, 20 µg/mL of coated S protein beads were pre-incubated with unlabeled nAbs diluted 1:2 in PBS for 40 min at RT. After incubation, the mix of Beads-antibody was washed with 100 µL of PBS-BSA 1%. Then, to analyze epitope competition, mAbs able to bind RBD Class 1/2, (J08), Class 3 (S309), and Class 4 (CR3022) were labeled with three different fluorophores (Alexa Fluor 647, 488, and 594) using Alexa Fluor NHS Ester kit (Thermo Scientific), were mixed and incubated with S protein-beads. Following 40 min of incubation at RT, the mix of Beads-antibodies was washed with PBS, resuspended in 150 µL of PBS-BSA 1%, and analyzed using BD LSR II flow cytometer (Becton Dickinson). Beads with or without S protein incubated with labeled

antibodies mix were used as a positive and negative control, respectively. FACSDiva Software (version 9) was used for data acquisition and analysis was performed using FlowJo (version 10).

## SARS-CoV-2 authentic viruses neutralization assay

All SARS-CoV-2 authentic virus neutralization assays were performed in the biosafety level 3 (BSL3) laboratories at Toscana Life Sciences in Siena (Italy) and Vismederi Srl, Siena (Italy). BSL3 laboratories are approved by a Certified Biosafety Professional and are inspected every year by local authorities. To evaluate the neutralization activity of identified nAbs against SARS-CoV-2 and all VoCs and evaluate the breadth of neutralization of this antibody is a cytopathic effect-based microneutralization assay (CPE-MN) was performed[3,4]. Briefly, the CPE-based neutralization assay sees the co-incubation of the antibody with a SARS-CoV-2 viral solution containing 100 median Tissue Culture Infectious Dose (100 $TCID_{50}$) of virus for 1 hour at 37 °C, 5% $CO_2$. The mixture was then added to the wells of a 96-well plate containing a sub-confluent Vero E6 (ATCC, Cat#CRL-1586) cell monolayer. Plates were incubated for 3–4 days at 37 °C in a humidified environment with 5% $CO_2$, then examined for CPE by means of an inverted optical microscope by two independent operators. All nAbs were tested at a starting dilution of 1:5 and the $IC_{100}$ was evaluated based on their initial concentration, while plasma samples were tested starting from a 1:10 dilution. Both nAbs and plasma samples were then diluted in step 1:2. Technical duplicates were performed for both nAbs and plasma samples. In each plate, positive and negative control were used as previously described[5].

## SARS-CoV-2 virus variants CPE-MN neutralization assay

The SARS-CoV-2 viruses used to perform the CPE-MN neutralization assay were the original Wuhan SARS-CoV-2 virus (SARS-CoV-2/INMI1-Isolate/2020/Italy: MT066156), SARS-CoV-2 B.1.1.7 (INMI GISAID accession number: EPI_ISL_736997), SARS-CoV-2 B.1.351 (EVAg Cod: 014V-04058), B.1.1.248 (EVAg CoD: 014V-04089), and B.1.617.2 (GISAID ID: EPI_ISL_2029113)[25].

## HEK293TN- hACE2 cell line generation

HEK293TN- hACE2 cell line was generated by lentiviral transduction of HEK293TN (System Bioscience, Cat#LV900A-1) cells as described in ref. 26. Lentiviral vectors were produced following a standard procedure based on calcium phosphate co-transfection with third-generation helper and transfer plasmids. The transfer vector pLENTI_hACE2_HygR was obtained by cloning of hACE2 from pcDNA3.1-hACE2 (Addgene #145033) into pLenti-CMV-GFP-Hygro (Addgene #17446). HEK293TN-hACE2 cells were maintained in DMEM, supplemented with 10% FBS, 1% glutamine, 1% penicillin/streptomycin, and 250 μg/ml hygromycin (GIBCO).

## Production of SARS-CoV-1 pseudoparticles

SARS-CoV-1 lentiviral pseudotype particles were generated as described in Conforti et al. for SARS-CoV-2[25] by using the SARS-CoV1 SPIKE plasmid pcDNA3.3_CoV1_D28 (Addgene plasmid # 170447).

## SARS-CoV-1 neutralization assay

For neutralization assay, HEK293TN-hACE2 cells were plated in white 96-well plates in a complete DMEM medium. 24 h later, cells were infected with 0.1 MOI of SARS-CoV-1 pseudoparticles that were previously incubated with serial dilution of purified or not purified (cell supernatant) mAb. In particular, a seven-point dose-response curve (plus PBS as untreated control), was obtained by diluting mAb or supernatant, respectively fivefold and threefold. Thereafter, nAbs of each dose-response curve point was added to the medium containing SARS-CoV-1 pseudoparticles adjusted to contain 0.1 MOI. After incubation for 1 h at 37 °C, 50 μl of mAb/SARS-CoV-1 pseudoparticles mixture was added to each well, and plates were incubated for 24 h at

37 °C. Each point was assayed in technical triplicates. After 24 h of incubation, cell infection was measured by luciferase assay using Bright-Glo™ Luciferase System (Promega) and Infinite F200 plate reader (Tecan) was used to read luminescence. Obtained relative light units (RLUs) were normalized to controls and dose-response curve were generated by nonlinear regression curve fitting with GraphPad Prism to calculate Neutralization Dose 50 ($ND_{50}$).

## Single-cell RT-PCR and Ig gene amplification and transcriptionally active PCR expression

To express our nAbs as full-length IgG1, 5 μL of cell lysate from the original 384-cell sorting plate were used for reverse transcription polymerase chain reaction (RT-PCR), and two rounds of PCRs (PCRI and PCRII-nested) as previously described[3,4]. Obtained PCRII products will be used to recover the antibody heavy and light chain sequences, through Sanger sequencing, and for antibody cloning into expression vectors as previously described[27–29]. Transcriptionally active PCR (TAP) reaction was performed using 5 μL of Q5 polymerase (NEB), 5 μL of GC Enhancer (NEB), 5 μL of 5X buffer,10 mM dNTPs, 0.125 μL of forward/reverse primers, and 3 μL of ligation product, using the following cycles: 98°/2′, 35 cycles 98°/10″, 61°/20″, 72°/1′, and 72°/5′. TAP products were purified under the same PCRII conditions, quantified by Qubit Fluorometric Quantitation assay (Invitrogen), and used for transient transfection in the Expi293F cell line following manufacturing instructions.

## Functional repertoire analyses

nAbs VH and VL sequence reads were manually curated and retrieved using a CLC sequence viewer (Qiagen). Aberrant sequences were removed from the data set. Analyzed reads were saved in FASTA format and the repertoire analyses was performed using Cloanalyst (http://www.bu.edu/computationalimmunology/research/software/)[30,31].

## Radar plot distribution of IGHV;IGHJ germlines

A radar plot was generated to display the distribution of IGHV;IGHJ germlines among the three nAbs groups: seronegative second dose, third dose, and seropositive second dose. Each star in the radar represents a particular IGHV;IGHJ germline combination, alphabetically sorted. Germline abundance is represented as a percentage of the total of each group. In addition, the log2 fold change for each combination of the three groups was calculated and represented as a concentric heatmap; the higher the fold change it is, the bigger is the first group with respect to the other, and vice versa. The figure was generated with R v4.1.1. and assembled with ggplot2 v3.3.5.

## Network plot of clonally-expanded antibody families

To investigate the genetic similarity within and between lineages, a network map was built by representing each clonal family with a centroid and connecting centroids sharing a similar sequence. The centroid sequence was computed with Cloanalyst to represent the average CDRH3 sequence for each clonal family, and Hamming distance was calculated for each antibody CDRH3 sequence to represent the relationship within the clonal family. Levenshtein distance was calculated between each centroid representative of each clonal family to investigate the relationship between clonal families. Levenshtein distance was calculated with the R package stringdist v0.9.8 (https://cran.r-project.org/web/packages/stringdist/index.html) and normalized between 0 and 1. A network graph was generated with the R package ggraph v2.0.5 (https://ggraph.data-imaginist.com/index.html) with Fruchterman–Reingold layout algorithm and the figure was assembled with ggplot2 v3.3.5. The size of the centroid is proportional to the number of antibodies belonging to the same clonal family, while the color of each node represents the antibody origin: light blue and dark blue for the seronegative second dose and seronegative third dose, respectively.

## Statistics and reproducibility

Statistical analysis was assessed with GraphPad Prism Version 8.0.2 (GraphPad Software, Inc., San Diego, CA). Nonparametric Mann–Whitney *t*-test was used to evaluate the statistical significance between the groups analyzed in this study. Statistical significance was shown as * for values ≤0.05, ** for values ≤0.01, *** for values ≤0.001, and **** for values ≤0.0001. No statistical methods were used to pre-determine the sample size of subjects and nAbs screened. All four subjects previously analyzed after two vaccine doses were re-enrolled after receiving the third dose. All isolated nAbs were analyzed in this study. No data were excluded from the analyses. The experiments were not randomized and investigators were not blinded to allocation during experiments and outcome assessment.

## Reporting summary

Further information on research design is available in the Nature Portfolio Reporting Summary linked to this article.

## Data availability

Source data are provided with this paper. All data supporting the findings in this study are available within the article or can be obtained from the corresponding author upon request. SARS-CoV-2 variant sequences were deposited and accessible from https://github.com/dasch-lab/SARS-CoV-2_nAb_third_dose Source data are provided with this paper.

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

## Acknowledgements

This work was funded by the European Research Council (ERC) advanced grant agreement number 787552 (vAMRes). This work was supported by a fundraising activity promoted by Unicoop Firenze, Coop Alleanza 3.0, Unicoop Tirreno, Coop Centro Italia, Coop Reno e Coop Amiatina. This publication was supported by the COVID-2020-12371817 project, which received funding from the Italian Ministry of Health. We would also like to acknowledge Dr. Jason McLellan, for kindly providing the S protein trimer, RBD, NTD, and S2 constructs, Dr. Olivier Schwartz, for providing the B.1.617.2 (delta) SARS-CoV-2 variant, and Dr. Piet Maes for providing the B.1.1.529.1 (BA.1) and B.1.1.529.2 (BA.2) SARS-CoV-2 variants. We would like to thank the nurse staff of the operative unit of the Department of Medical Sciences, Infectious and Tropical Diseases Unit, Siena University Hospital, Siena, Italy, and all the COVID-19 vaccinated donors for participating in this study.

## Author contributions

Conceived the study: E.A. and R.R.; Enrolled COVID-19 vaccinees: F.M., M.F., I.R., and M.T.; Performed PBMC isolation and single-cell sorting:

E.A. and I.P.; Performed ELISAs and competition assays: I.P., V.A., and G.A.; Recovered nAbs VH and VL and expressed antibodies: I.P. and N.M.; Recovered VH and VL sequences and performed the repertoire analyses: P.P., E.A., and G.M.; Produced and purified SARS-CoV-2 S protein constructs: E.P.; Performed neutralization assays in BSL3 facilities: E.A., G. Pie., G.Pic., M.L., L.B., and G.G.; Performed SARS-CoV-1 pseudotype neutralization assays: S.M. and L.D.; Supported day-by-day laboratory activities and management: C.D.S.; Manuscript writing: E.A. and R.R.; Final revision of the manuscript: E.A., I.P., G.Pie., G.Pic., V.A., G.A., P.P., N.M., E.P., L.B., G.G., M.L., S.M., L.D., C.D.S., M.F., I.R., M.T., F.M., C.S., R.D.F., E.M., and R.R.; Coordinated the project: E.A., C.S., E.M., R.D.F., and R.R.

## Competing interests

E.A., I.P., N.M., P.P., E.P., V.A., C.D.S., C.S., and R.R. are listed as inventors of full-length human monoclonal antibodies described in Italian patent applications n. 102020000015754 filed on June 30, 2020, 102020000018955 filed on August 3, 2020, and 102020000029969 filed on 4 December 2020, and the international patent system number PCT/IB2021/055755 filed on 28 June 2021. All patents were submitted by Fondazione Toscana Life Sciences, Siena, Italy. R.D.F. is a consultant for Moderna on activities not related to SARS-CoV-2. The remaining authors declare no competing interests.
