## [Peer Review File · Nature Communications]

B cell analyses of COVID-19 mRNA third vaccination reveals a unique, hybrid immunity-like, antibody responseREVIEWER COMMENTS

Reviewer #1 (Remarks to the Author):

Three background papers inform the review of this latest manuscript from Andreano et al.

<https://pubmed.ncbi.nlm.nih.gov/11713261/>

Liang et al TAP

<https://www.sciencedirect.com/science/article/pii/S0092867421002245?via=ihub>

Rappouli 1 Cell

<https://www.nature.com/articles/s41586-021-04117-7>

Rappouli 2 Nat Comm

The first paper describes a method called Transcriptional Active PCR (TAP) that is put to good use. TAP is proven here to be an efficient way to generate thousands of individual monoclonal Abs derived from single B cells. The Cell paper reports on the generation 4000 individual mAbs from a few convalescent COVID cases that recognize Spike from the Wuhan strain. 10% (~400) are neutralizing and 1% (~40) have potent neutralizing activity. The potent mAbs show prophylactic and therapeutic efficacy in the hamster model. This was followed by the Nature Communications reporting isolation of 276 neutralizing mAbs from from volunteers vaccinated with the BioNTech mRNA vaccine. Among the 276 neutralizing Abs there was >1000 fold range in their neutralizing potency against the Wuhan strain. In the vaccinated people who had recovered from a natural exposure only 20% of these Abs could neutralize the Omicron viruses BA1 and BA2. In the vaccinated people who did not have a previous infection, <10% of their Abs could neutralize Omicron.

The new manuscript is a logical extension of these initial studies to measure cross reactive neutralizing Ab resulting from a third vaccination. 4000 anti-spike mAbs were expressed and a collection of 350 were discovered with neutralizing activity. Neutralizing activity of the mAbs in this collection varied by 3 orders of magnitude. There was more cross neutralizing activity evident (~50%) against alpha, beta, gamma, and delta but Omicron remained at about 20% after the third immunization.

The basic understanding of adaptive immunity revealed from this series of papers is eye popping and teaches immunology,

- Number of different clonal B cells that can be found directed against the same antigen
- Breadth of neutralizing efficacy within this collection of neutralizing Abs, from very weak to very strong
- Another metric to understand class switching, affinity maturation and epitope selection

This is a well written manuscript, with good design of experiments that include appropriate controls and sufficient data collection. The central point of this manuscript is that seronegative (previously uninfected) people receiving three doses (SN3) of the Covid vaccine induces a better antibody response (higher neutralizing Ab titers, more cross-reactivity to antigen variants, more hypermutated antibodies, and antibodies that come from a more diverse number of variable chain antibody gene families) compared to cohorts receiving 2 doses (SN2) of the Covid vaccine. In addition, the SN3 response is more similar to the recent concept of hybrid immunity, i.e. immunity in previously infected (seropositive) people receiving 2 doses of vaccine (cohort SP2).

Although the data are robust, and while clearly the central issue of this paper is the similarities and differences between the antibody response of SN3 vs SN2, perhaps better

discussion of how SN3 and SP2 compare can be developed. The reason is that while previously 2 doses of the vaccine were considered to render the subject 'fully vaccinated' this definition at this moment refers to 3 doses of the vaccine. More relevant for comparison purposes is the fact that while SN3 react well against the original Wuhan strain, SN3 still react less well to the omicron strains at the same 20% level as the previous publication.

The major conundrum with this manuscript is that the SN3 group is similar to the SP2 group, whereas hybrid immunity (i.e. SP2) is generally considered to be stronger and more diverse in the makeup of activated lymphocyte clones (Crotty, S. 2021. *Science*. 372: 1392-1393). Crotty writes that in previously infected individuals who are later vaccinated as "impressive synergy occurs—a "hybrid vigor immunity" resulting from a combination of natural immunity and vaccine-generated immunity". How one defines hybrid immunity – the number of vaccine doses, whether they were administered before or after infection, and how symptomatic was the infection – all affect hybrid immunity. Thus, it could be that hybrid immunity is most pronounced when comparing individuals receiving 2 vaccine doses to those who receive only 1 dose but are infected, whereas in the current manuscript the emphasis is on SN3 vs SP2 (and both vs SN2). Although hybrid immunity is a recent and understudied topic, the authors have not addressed their findings in this broader background in the Discussion section. The authors have focused in the Discussion on the minutia of their data showing the superior response in SN3 compared to SN2, but they may also discuss how repeated boosts affect vaccine performance towards mutating variants.

Minor points include providing a reference for certain techniques such as transcriptionally active PCR (TAP) (line 405). Since detecting antigen-specific B cells with fluorescently labeled antigens is such a key element of this manuscript, the description of how this was undertaken should be made more explicit in the Methods section. In particular, it's likely that the authors used SARS-CoV-2 S antigen that was genetically engineered to express the Strep-tag II or Twin-Strep-tag in order for it to then bind fluorescent Strep-Tactin, though this information is not provided (lines 298-299). Furthermore, although memory B cells (MBC) were defined elsewhere in the manuscript as CD19+ CD27+ IgD- IgM- (line 72), and while the antibodies used to sort antigen-specific and total MBCs have been listed in the Methods section (lines 293-305), it would be clearer to provide the entire gating strategy.

Reviewer #2 (Remarks to the Author):

Andreano et al, evaluated antibody response using Mabs following vaccination against Omicron BA1 and BA2 variants.

The study demonstrates the 3rd vaccination induces a better diverse nAbs than post-2nd vaccination antibody response. Authors evaluated neutralization against Omicron BA1 and BA2 following 2nd and post-3rd vaccination and competition-based classification as well as IgG sequencing.

This study builds on previous study by the same group and others. It has some interesting aspects. Addressing the comments below will significantly improve the manuscript as well as make it relevant to the ongoing pandemic.

MAJOR COMMENTS:

1. The study measures neutralization against Omicron BA.1 and BA.2 strains. It will be important to analyze the neutralizing capacity of these MAbs with currently circulating predominant Omicron subvariants BA.2.12.1, BA.3 and BA.4/BA.5 to understand the importance of nAbs induced by post-2nd, post-3rd vaccination so as to make this study

relevant to the current state of the COVID-19 pandemic.

2. Several studies show that 3rd vaccine dose primarily recalls the memory response induced by earlier primary series of vaccination. Why the gene usage in Abs post-3rd vx is different from post 1/2 vx dose? Why not 3rd vx predominantly recall of immune response by previous 1/2 vaccination dose? It should be discussed in context of earlier literature.

3. Authors mention that no nAbs were identified in S2 domain (line 85). Some previous studies have demonstrated nAbs targeting the S2 domain using classical PRNT assay. S2 is conserved between SARS-CoV-2 and other seasonal hCoVs. What is the antibody target site on S2 (heptad repeat etc)? Can authors provide the total number of S2 specific Mabs and test them in classical PRNT to be accurate?

4. The neutralization activity and gene-usage different between Class 1/2 nAbs and between SN3 vs SP2 for the Omicron nAbs. Can authors discuss potential mechanism for these observations? How is this happening in immune-exposed individuals? Where are these naïve B cells coming from and why they were not induced by post-1/2 vaccination?

5. Is there any correlation between neutralization breadth of nAbs and time interval between 2-3rd vaccine dose in these individuals?

Reviewer #3 (Remarks to the Author):

This is an interesting, concisely described study and a comprehensive analysis of the spike antibody repertoire following a third mRNA vaccine dose compared with the second.

As the authors note from their referenced studies in Cell and Nature, similar findings have been described elsewhere. This analysis offers additional detail, but is of limited significance. For example the finding of limited cross-reactivity of antibodies to SARS-CoV-1 post-3rd dose has been described previously (if not in such a granular and precise way) and is not unexpected.

A few minor points:

- a summary (e.g. in extended Data Table 1) of the seropositive cohort would be helpful for readers.
- how was intervening infection excluded? I assume no infections were reported, but was an asymptomatic infection screened for by e.g. testing anti-N?
- while n=3:1, was there any evidence of a different antibody response among patients who received the Moderna versus Pfizer vaccines?
- is there capacity to include BA4/5 analysis, given the significance of these sub-variants currently

REBUTTAL: NCOMMS-22-20749-T - COVID-19 mRNA third dose induces a unique hybrid immunity-like antibody response

General comment

We thank the editor and reviewers for their critical assessment of our work and for the positive feedbacks herein provided. In the revised version of the manuscript, we amended the text and table in accordance with the reviewers' comments. In addition, you can find below a point-by-point response to all comments.

REVIEWER COMMENTS

Reviewer #1 (Remarks to the Author):

Three background papers inform the review of this latest manuscript from Andreato et al.

<https://pubmed.ncbi.nlm.nih.gov/11713261/>

Liang et al TAP

<https://www.sciencedirect.com/science/article/pii/S0092867421002245?via=ihub>

Rappouli 1 Cell

<https://www.nature.com/articles/s41586-021-04117-7>

Rappouli 2 Nat Comm

The first paper describes a method called Transcriptional Active PCR (TAP) that is put to good use. TAP is proven here to be an efficient way to generate thousands of individual monoclonal Abs derived from single B cells. The Cell paper reports on the generation 4000 individual mAbs from a few convalescent COVID cases that recognize Spike from the Wuhan strain. 10% (~400) are neutralizing and 1% (~40) have potent neutralizing activity. The potent mAbs show prophylactic and therapeutic efficacy in the hamster model. This was followed by the Nature Communications reporting isolation of 276 neutralizing mAbs from from volunteers vaccinated with the BioNTech mRNA vaccine. Among the 276 neutralizing Abs there was >1000 fold range in their neutralizing potency against the Wuhan strain. In the vaccinated people who had recovered from a natural exposure only 20% of these Abs could neutralize the Omicron viruses BA1 and BA2. In the vaccinated people who did not have a previous infection, <10% of their Abs could neutralize Omicron.

The new manuscript is a logical extension of these initial studies to measure cross reactive neutralizing Ab resulting from a third vaccination. 4000 anti-spike mAbs were expressed and a collection of 350 were discovered with neutralizing activity. Neutralizing activity of the mAbs in this collection varied by 3 orders of magnitude. There was more cross neutralizing activity evident (~50%) against alpha, beta, gamma, and delta but Omicron remained at about 20% after the third immunization.

The basic understanding of adaptive immunity revealed from this series of papers is eye popping and teaches immunology,

- Number of different clonal B cells that can be found directed against the same antigen
- Breadth of neutralizing efficacy within this collection of neutralizing Abs, from very weak to very strong
- Another metric to understand class switching, affinity maturation and epitope selection

This is a well written manuscript, with good design of experiments that include appropriate controls and sufficient data collection. The central point of this manuscript is that seronegative (previously uninfected) people receiving three doses (SN3) of the Covid vaccine induces a better antibody response (higher neutralizing Ab titers, more cross-reactivity to antigen variants, more hypermutated antibodies, and antibodies that come from a more diverse number of variable chain antibody gene families) compared to cohorts receiving 2 doses (SN2) of the Covid vaccine. In addition, the SN3 response is more similar to the

recent concept of hybrid immunity, i.e. immunity in previously infected (seropositive) people receiving 2 doses of vaccine (cohort SP2).

R1: Although the data are robust, and while clearly the central issue of this paper is the similarities and differences between the antibody response of SN3 vs SN2, perhaps better discussion of how SN3 and SP2 compare can be developed. The reason is that while previously 2 doses of the vaccine were considered to render the subject ‘fully vaccinated’ this definition at this moment refers to 3 doses of the vaccine. More relevant for comparison purposes is the fact that while SN3 react well against the original Wuhan strain, SN3 still react less well to the omicron strains at the same 20% level as the previous publication.

A1: We thank the reviewer for this comment. As rightfully pointed out, our work focuses mainly on the comparison between SN2 and SN3 to unravel the longitudinal evolution of the functional antibody response and B cell repertoire. Nevertheless, throughout the text several comparisons between SN3 and SP2 were made. Indeed, the overall nAb neutralization potency (reported as GM-IC₁₀₀) (Extended Data Figure 3), and functional antibody repertoire analyses (Figure 3A and Extended Data Figure 6) were also compared between SN3 and SP2. In addition, to better understand similarities and differences of the antibody response mounted after three doses and with “hybrid immunity”, an in-depth analysis of nAb classes of SARS-CoV-2 variants was performed exclusively between SN3 and SP2 (Figure 2d-e). These points were also thoroughly highlighted in the Discussion section page 8, line 162-170, 187-191 of the revised manuscript version. However, we have further expanded the discussion between SN3 and SP2 which can be found at page 8, line 192 – 198 of the revised manuscript.

R2: The major conundrum with this manuscript is that the SN3 group is similar to the SP2 group, whereas hybrid immunity (i.e. SP2) is generally considered to be stronger and more diverse in the makeup of activated lymphocyte clones (Crotty, S. 2021. *Science*. 372: 1392-1393). Crotty writes that in previously infected individuals who are later vaccinated as “impressive synergy occurs—a “hybrid vigor immunity” resulting from a combination of natural immunity and vaccine-generated immunity”. How one defines hybrid immunity – the number of vaccine doses, whether they were administered before or after infection, and how symptomatic was the infection – all affect hybrid immunity. Thus, it could be that hybrid immunity is most pronounced when comparing individuals receiving 2 vaccine doses to those who receive only 1 dose but are infected, whereas in the current manuscript the emphasis is on SN3 vs SP2 (and both vs SN2). Although hybrid immunity is a recent and understudied topic, the authors have not addressed their findings in this broader background in the Discussion section. The authors have focused in the Discussion on the minutia of their data showing the superior response in SN3 compared to SN2, but they may also discuss how repeated boosts affect vaccine performance towards mutating variants.

A2: We agree with this reviewer that the “hybrid immunity” topic remains understudied, and several combinations of vaccination and infection (with different SARS-CoV-2 variants) make this immune state extremely heterogenous. Despite this observation, the authors believe that the conundrum with the similarity in SARS-CoV-2 neutralization potency and breadth between SN3 and SP2 can be explained with the following points. The first point is that the concept of hybrid immunity introduced by Prof. Crotty in June 2021 (<https://doi.org/10.1126/science.abj2258>), was published few months before the recommendation of the third vaccine dose. Therefore, the studies on which the concept of hybrid immunity was based were performed exclusively on subjects with one or two vaccine doses compared to vaccinated individuals with previous infection (<https://doi.org/10.1126/science.abg9175>; <https://doi.org/10.1126/science.abh1282>). As a result, the initial concept of hybrid immunity did not consider the full vaccination schedule as three doses and could have underestimated the power of a third booster dose. The second point is that recent studies compare three vaccine doses with hybrid immunity based on vaccination plus omicron infection which display a higher ability to cover all the different SARS-CoV-2 omicron subvariants (<https://doi.org/10.1056/NEJMc2206576>; <https://doi.org/10.1016/j.cell.2022.06.005>). Conversely, the SP2

cohort analyzed in our work is extremely homogenous. Indeed, all subjects were infected in October – November 2020 (<https://doi.org/10.1038/s41586-021-04117-7>), when only the D614G variant was circulating worldwide, and vaccinated with the same S protein antigen encoded by the BNT162b2 mRNA vaccine. Hence, our SN3 and SP2 cohorts were primed, through vaccination and infection respectively, and boosted with the same S protein. The single cell analyses performed on such homogenous populations provide unique insights on how infection and vaccination can drive an antibody response with analogous potency and breadth but based on the expansion of different B cell germ lines induced by virions and vaccines which display the same S protein antigen.

To conclude, thanks to the data provided in our work, we can state that repeated vaccine boosts are fundamental and induce a unique hybrid immunity-like antibody response able to better protect all individuals against current and emerging SARS-CoV-2 variants.

R3: Minor points include providing a reference for certain techniques such as transcriptionally active PCR (TAP) (line 405). Since detecting antigen-specific B cells with fluorescently labeled antigens is such a key element of this manuscript, the description of how this was undertaken should be made more explicit in the Methods section. In particular, it's likely that the authors used SARS-CoV-2 S antigen that was genetically engineered to express the Strep-tag II or Twin-Strep-tag in order for it to then bind fluorescent Strep-Tactin, though this information is not provided (lines 298-299). Furthermore, although memory B cells (MBC) were defined elsewhere in the manuscript as CD19⁺ CD27⁺ IgD⁻ IgM⁻ (line 72), and while the antibodies used to sort antigen-specific and total MBCs have been listed in the Methods section (lines 293-305), it would be clearer to provide the entire gating strategy.

A3: Reference by Liang et al describing the TAP approach was added at line 328 (reference 31). Information about SARS-CoV-2 spike protein used for single cell sorting and gating strategy have been added in the "Single cell sorting of SARS-CoV-2 S protein⁺ memory B cells from COVID-19 vaccinees" paragraph at page 10, line 219 – 222 and 226 respectively of the revised manuscript.

Reviewer #2 (Remarks to the Author):

Andreano et al, evaluated antibody response using MAbs following vaccination against Omicron BA1 and BA2 variants. The study demonstrates the 3rd vaccination induces a better diverse nAbs than post-2nd vaccination antibody response. Authors evaluated neutralization against Omicron BA1 and BA2 following 2nd and post-3rd vaccination and competition-based classification as well as IgG sequencing. This study builds on previous study by the same group and others. It has some interesting aspects. Addressing the comments below will significantly improve the manuscript as well as make it relevant to the ongoing pandemic.

MAJOR COMMENTS:

R1: The study measures neutralization against Omicron BA.1 and BA.2 strains. It will be important to analyze the neutralizing capacity of these MAbs with currently circulating predominant Omicron subvariants BA.2.12.1, BA.3 and BA.4/BA.5 to understand the importance of nAbs induced by post-2nd, post-3rd vaccination so as to make this study relevant to the current state of the COVID-19 pandemic.

A1: We thank the reviewer for this comment, and we agree that the current state of the COVID-19 pandemic shows a different viral epidemiology compared to when this manuscript was initially submitted. However, given the pace at which new variants evolve, it is impossible to obtain data on the newest variants.

R2: Several studies show that 3rd vaccine dose primarily recalls the memory response induced by earlier primary series of vaccination. Why the gene usage in Abs post-3rd vx is different from post 1/2 vx dose? Why not 3rd vx predominantly recall of immune response by previous 1/2 vaccination dose? It should be discussed in context of earlier literature.

A2: A recent study published in Nature in April 2022 (<https://doi.org/10.1038/s41586-022-04778-y>), following the analyses RBD-binding memory B cells, also showed that a third booster dose leads to the expansion of memory B cells present after the second dose as well as the emergence of new clones. We observed that also neutralizing antibodies derive from new clones not expanded during the first or second dose. In our work (Figure 3c-f) we demonstrated that emerging clones present much higher V gene somatic mutation levels compared to the same germline after two vaccine doses. Therefore, we can postulate that new clones needed further maturation, induced by a third booster dose, before acquiring high level of functionality and being selected for further expansion. This comment has been added into the Discussion section of this manuscript at page 8 line 178 – 180 of the revised manuscript.

R3: Authors mention that no nAbs were identified in S2 domain (line 85). Some previous studies have demonstrated nAbs targeting the S2 domain using classical PRNT assay. S2 is conserved between SARS-CoV-2 and other seasonal hCoVs. What is the antibody target site on S2 (heptad repeat etc)? Can authors provide the total number of S2 specific MAbs and test them in classical PRNT to be accurate?

A3: With our cytopathic effect microneutralization assay (CPE-MN) we previously showed to be able to identify S2-targeting nAbs (<https://doi.org/10.1016/j.cell.2021.02.035>). Therefore, we already demonstrated that our method is sufficiently sensitive to identify S2-targeting nAbs which were shown in several works, including ours, to have low neutralization activity.

R4: The neutralization activity and gene-usage different between Class 1/2 nAbs and between SN3 vs SP2 for the Omicron nAbs. Can authors discuss potential mechanism for these observations? How is this happening in immune-exposed individuals? Where are these naïve B cells coming from and why they were not induced by post-1/2 vaccination?

A4: As pointed out in our Discussion section, we believe that natural infection and endogenous production of virions presenting the wild type spike protein can expose different portions of the RBD compared to the stabilized pre-fusion SARS-CoV-2 S protein (S-2P) encoded by mRNA vaccines. Indeed, S-2P was shown to be

mainly locked in the one RBD-up conformation (<https://doi.org/10.1126/science.abb2507>; <https://doi.org/10.1126/science.abd0826>), while the wild type form of the S protein is a metastable glycoprotein which shows one or two RBD-up, and can transiently switch from a pre-fusion to a post-fusion conformation independently from binding to the targeted cells (<https://doi.org/10.7554/eLife.75720>; <https://doi.org/10.1126/science.abd4251>). The differences between the wild type protein and the stabilized S protein can make available different epitope regions which in turn could drive the expansion of different B cell germlines.

As for the difference of B cells between SN2 and SN3, we would like to specify that we were not discussing on naïve B cells. Indeed, we assume that antibodies encoded by specific B cell germlines (like IGHV1-58 and IGHV1-69), despite initially recruited by the first and second vaccination dose, were not mature enough to neutralize the virus. Conversely, the third dose enhanced the affinity maturation of these B cell germlines which become highly functional and were consequently expanded to mount a protective antibody response.

R5: Is there any correlation between neutralization breadth of nAbs and time interval between 2-3rd vaccine dose in these individuals?

A5: As shown in Extended Data Table 2, no correlation is observed between neutralization breadth and time interval between second and third vaccine dose for the four subjects evaluated in this study.

Reviewer #3 (Remarks to the Author):

This is an interesting, concisely described study and a comprehensive analysis of the spike antibody repertoire following a third mRNA vaccine dose compared with the second.

As the authors note from their referenced studies in Cell and Nature, similar findings have been described elsewhere. This analysis offers additional detail, but is of limited significance. For example the finding of limited cross-reactivity of antibodies to SARS-CoV-1 post-3rd dose has been described previously (if not in such a granular and precise way) and is not unexpected.

A few minor points:

R1: A summary (e.g. in extended Data Table 1) of the seropositive cohort would be helpful for readers.

A1: Extended Data Table 1 was modified to include also the seropositive cohort.

R2: How was intervening infection excluded? I assume no infections were reported, but was an asymptomatic infection screened for by e.g. testing anti-N?

A2: Infection was excluded from SN3 as no SARS-CoV-2 infection was reported. Despite anti-N screening was not performed, our neutralizing antibodies repertoire analyses reported in Extended Data Figure 6, showed extremely similar profiles among subjects, further suggesting that subjects were not exposed to SARS-CoV-2 infection.

R3: While n=3:1, was there any evidence of a different antibody response among patients who received the Moderna versus Pfizer vaccines?

A3: Major differences were not observed between the three subjects (VAC-001, VAC-002 and VAC-008) which received the Pfizer (BNT162b2) vaccine and the single subject (VAC-010) which was immunized with the Moderna (mRNA-1273) vaccine. Extended Data Table 2 and Extended Data Figure 6 show no differences between the three Pfizer and the one Moderna vaccinees in frequency of neutralizing antibodies and B cell repertoire respectively.

R4: Is there capacity to include BA4/5 analysis, given the significance of these sub-variants currently

A4: We thank the reviewer for this comment, and we agree that the current state of the COVID-19 pandemic shows a different viral epidemiology compared to when this manuscript was initially submitted. However, given the pace at which new variants evolve, it is impossible to obtain data on the newest variants.

REVIEWERS' COMMENTS

Reviewer #1 (Remarks to the Author):

The authors responded well to my comments.

Reviewer #3 (Remarks to the Author):

Reviewer comments have been addressed adequately. I have nil further to add.

Point-by-point response: NCOMMS-22-20749A - B cell analyses of COVID-19 mRNA third vaccination reveals a unique, hybrid immunity-like, antibody response

General comment

We thank the editor and reviewers for their critical assessment of our work and for the positive feedbacks herein provided. In the revised version of the manuscript, we amended the text and figures in accordance with the journal policies and editors' comments. In addition, you can find below a point-by-point response to the referees' comments.

REVIEWER COMMENTS

Reviewer #1 (Remarks to the Author):

R1: The authors responded well to my comments.

A1: We thank this reviewer for the assessment of our work and for this final comment.

Reviewer #3 (Remarks to the Author):

R1: Reviewer comments have been addressed adequately. I have nil further to add.

A1: We thank this reviewer for the assessment of our work and for this final comment.